# Contactless Technologies, Sensors, and Systems for Cardiac and Respiratory Measurement during Sleep: A Systematic Review

**DOI:** 10.3390/s23115038

**Published:** 2023-05-24

**Authors:** Andrei Boiko, Natividad Martínez Madrid, Ralf Seepold

**Affiliations:** 1Ubiquitous Computing Laboratory, Department of Computer Science, HTWG Konstanz—University of Applied Sciences, Alfred-Wachtel-Str. 8, 78462 Konstanz, Germany; ralf.seepold@htwg-konstanz.de; 2Internet of Things Laboratory, School of Informatics, Reutlingen University, Alteburgstr. 150, 72762 Reutlingen, Germany; natividad.martinez@reutlingen-unversity.de

**Keywords:** contactless technologies, sensors, cardiac activity, respiratory activity, sleep measurements, health monitoring systems, sleep monitoring systems

## Abstract

Sleep is essential to physical and mental health. However, the traditional approach to sleep analysis—polysomnography (PSG)—is intrusive and expensive. Therefore, there is great interest in the development of non-contact, non-invasive, and non-intrusive sleep monitoring systems and technologies that can reliably and accurately measure cardiorespiratory parameters with minimal impact on the patient. This has led to the development of other relevant approaches, which are characterised, for example, by the fact that they allow greater freedom of movement and do not require direct contact with the body, i.e., they are non-contact. This systematic review discusses the relevant methods and technologies for non-contact monitoring of cardiorespiratory activity during sleep. Taking into account the current state of the art in non-intrusive technologies, we can identify the methods of non-intrusive monitoring of cardiac and respiratory activity, the technologies and types of sensors used, and the possible physiological parameters available for analysis. To do this, we conducted a literature review and summarised current research on the use of non-contact technologies for non-intrusive monitoring of cardiac and respiratory activity. The inclusion and exclusion criteria for the selection of publications were established prior to the start of the search. Publications were assessed using one main question and several specific questions. We obtained 3774 unique articles from four literature databases (Web of Science, IEEE Xplore, PubMed, and Scopus) and checked them for relevance, resulting in 54 articles that were analysed in a structured way using terminology. The result was 15 different types of sensors and devices (e.g., radar, temperature sensors, motion sensors, cameras) that can be installed in hospital wards and departments or in the environment. The ability to detect heart rate, respiratory rate, and sleep disorders such as apnoea was among the characteristics examined to investigate the overall effectiveness of the systems and technologies considered for cardiorespiratory monitoring. In addition, the advantages and disadvantages of the considered systems and technologies were identified by answering the identified research questions. The results obtained allow us to determine the current trends and the vector of development of medical technologies in sleep medicine for future researchers and research.

## 1. Introduction

Healthcare systems around the world are facing significant challenges such as a rapidly ageing population, increasing numbers of people with chronic and infectious diseases, rising costs, and inefficient healthcare systems. Healthcare providers are looking for new non-invasive solutions that can improve the healthcare experience for patients while maintaining the cost of the services provided [1]. At the same time, continuous monitoring of a patient’s vital signs is essential for many chronic diseases. This can be performed both at home and in hospital. However, with the use of existing accepted contact-based monitoring systems and technologies, the intrusiveness of such monitoring into a patient’s life often becomes a barrier to living a normal life. The development and practical application of a non-contact monitoring system can help to overcome the above problem. This can be achieved by installing a system in a patient’s bed in a hospital, or even in an ordinary bed at home, allowing vital signs to be monitored anywhere [2,3]. Meanwhile, possible changes in a patient’s health could be detected by monitoring heart and respiratory rates, which would allow these vital signals to be defined as essential. Therefore, the application of non-contact technologies for monitoring a patient’s cardiorespiratory activity can play a significant role in improving the quality of life of the population.

In addition to the above, sleep is crucial for a healthy life and wellbeing. Improving the quality of sleep is essential, and sleep disorders need to be identified and treated promptly because they can lead to other health problems [4]. For example, poor sleep quality or sleep disorders lead to daytime fatigue, which impairs the mental and physical quality of daytime activities and increases the risk of accidents [5]. Monitoring sleep includes monitoring of vital signs. The latter is particularly important in view of the increasing risk of cardiovascular disease. Therefore, the need and importance of monitoring vital signs—including sleep quality, cardiac activity, and respiratory activity—becomes extremely relevant [6].

The gold standard for diagnosing and monitoring human vital signs—particularly during sleep—is the polysomnography (PSG) monitoring system, which records the electrical potentials of the brain and heart, eye movements, muscle activity, respiratory effort, airflow, oxygen saturation, and leg movements during the night [7]. Despite the quality and reliability of the PSG system, it is not well suited to long-term continuous monitoring, due to limited mobility, and it can cause irritation, anxiety, and discomfort to patients during monitoring [8]. For this reason, there has been increased interest in recent years in the development of commercial and research-oriented non-invasive and contactless sleep measurement devices, which have shown promising results in the detection and measurement of vital signs and sleep events [9]. These limitations have led to an increased demand for non-contact sleep monitoring systems.

As described in the literature, the problem with obtaining accurate readings from a non-contact monitoring device can be related to, for example, background interference, motion artefacts, and electromagnetic interference. In addition, for continuous performance monitoring in particular, problems are associated with the complexity of the sleep environment, noise associated with unpredictable body movements, body orientation, changes in sleep posture, multiple subject suppression, unwanted harmonics, and intermodulation [10]. Furthermore, there have been numerous reviews, comparative studies [11,12,13], and the development of intelligent systems [14] for unobtrusive [15] and non-contact monitoring of physiological vital signs for sleep monitoring. However, for example, in the work of Savage et al. [12], the study is presented at an early stage and, despite impressive results, requires an expansion of subjects. Ren et al. [11], when developing a radar-based system, found that the results remained acceptable at a small distance between the patient and the radar, which could affect aspects such as the unobtrusiveness of the system. A system based on a microbend fibre-optic sensor mat [15] is expensive to maintain, despite its high accuracy in determining physiological parameters. In addition to the above methods, non-invasive registration of cardiac [16] and respiratory [17] signals is a hot topic of research, which has been made possible, for example, by ballistocardiography (BCG) [16]. BCG is currently generating a lot of interest because of the revolution in information technology, including hardware, software and services. We can embed BCG sensors in the environment without the need for medical personnel to be present. Consequently, this has a significant impact on existing e-health systems [1]. In [18], a new approach to sleep stage classification is presented using a low-cost and non-contact fusion of multimodal sensors that extract sleep-related vital signs from radar signals and sound-based context awareness techniques. However, as the authors noted, there was instability in the continuous operation and recording of data from one of the sensors, resulting in the loss of important information. Another limitation of this system was low performance in some modes of operation. There are also approaches based on piezoelectric sensors [19], acoustic signal analysis [20], the combination of heart rate and motion variability [21], etc. According to the authors, this last approach [21] requires an extension of the study and of the algorithm for the determination. Sleep apnoea—a cardiorespiratory monitoring problem—can also be detected non-invasively, as presented for example in [8]. However, all of these approaches are still under development, and their applicability is limited.

This paper reviews the current state of contactless and non-intrusive technologies and approaches for cardiorespiratory monitoring. This includes a review of existing monitoring methods and techniques, technologies for their implementation, and the types of sensors used to implement cardiorespiratory monitoring. In addition, this article also discusses the medical applications that may exist through the use of the technology. Finally, this review includes a comparison of the effectiveness of contact and non-contact technologies for monitoring cardiac and respiratory activity during sleep. The results obtained make it possible to determine the current trends and the vector of development of medical technologies in the field of monitoring physiological parameters of vital signals in sleep medicine for future researchers and investigations.

Section 2 details the methodology used in this study, including the inclusion/exclusion criteria for publications, keywords used to search for articles, and a description of the search process. Section 3 details the findings, including the answers to the research questions. We highlight the discussion of the findings in Section 4 and draw conclusions in Section 5.

### 1.1. Review Questions

#### 1.1.1. Main Review Question (MRQ)

How can cardiac activity and respiration be contactlessly monitored during sleep?

#### 1.1.2. Specific Review Questions (SRQs)

Which technologies can be used for contactless measurement of cardiac activity and respiration during sleep?Which sensors are used for those technologies?Which physiological parameters can be extracted out of those sensors?What are the medical applications of contactless cardiac and respiratory monitoring during sleep?What are the differences in the quality of the measurements (contactless vs. contact-based/attached devices)?

The criteria for inclusion and exclusion of papers and database searches are mainly explained in the Materials and Methods section. In the Results section, we analyse all of the publications that meet the criteria, focusing on the number of physiological signals used and the precision in detecting vital signs and physiological parameters corresponding to cardiorespiratory activity—heart rate (HR) and respiratory rate (RR). At the same time, we note that the detection of physiological parameters of other vital signals (e.g., blood pressure or temperature) is also improved. Finally, the Discussion and Conclusions include a general interpretation of the results and the main information extracted from the analysis.

## 2. Materials and Methods

This review was conducted in accordance with “The PRISMA 2020 statement”—an update of the PRISMA (Preferred Reporting Items for Systematic Reviews and Meta-analyses) guidelines [22].

### 2.1. Eligibility Criteria

It is important to note that publications were eligible for the study based on their specific inclusion/exclusion criteria. Thus, the selection criteria relate not only to the type of publication and the relevance of the topic, but also to the details of the experiment to validate the monitoring system or technology (e.g., number of subjects and physiological parameters measured). Consequently, the number of publications for analysis was reduced to a minimum. Below are the inclusion and exclusion options that we defined for the publications.

The inclusion criteria were as follows:Study type: randomised controlled and clinical trials; results of testing the required systems and technologies at home, in labs, or in nursing houses; research journal articles and conference publications;Population: subjects who participated in trials measuring cardiac or respiratory movements during sleep;The document must have been published in a peer-reviewed format;The approach and system should have been implemented and tested at least at the level of a prototype;The system should have been developed for human measures (not for animals);The system should have been developed to fulfil at least one of the following aspects:
○Measuring cardiac parameters, heart rate, etc.;○Measuring respiratory rate from the contactless sensor/technology meeting the requirements mentioned above;○The system was developed to monitor sleep quality;○The system was developed to measure or verify a breathing- or cardiac-related disease;○The system was developed to measure or verify one of the diseases related to sleep;The method of data transmission or processing does not affect the inclusion/exclusion.The exclusion criteria were as follows:Book chapters, white papers, editorials, and perspectives;Papers not written in English;The paper was excluded if only the concept had been presented without any validation;The paper was excluded if only using wearable devices;Studies not related to measuring vital sign parameters or aforementioned (breathing) movements during sleep;Studies with publication dates older than five years when the systematic review was performed (2017–2022);Studies that are not focused on the use of contactless systems/methods/technologies for measuring the above parameters;Studies that included fewer than five subjects in an experiment;Published data not available.

### 2.2. Search Strategy and Information Sources

The search string that we developed reflects three aspects (see the entire string in Appendix A):

Activity, the target group, and measures comprise the terms on physiological characteristics/parameters for measurement or area where these parameters are required, such as “breathing rate”, “respiratory rate”, “heart rate”, “vital sign”, “cardiorespiratory”, “cardiovascular”, and “sleep”.

Technology, sensor, system to detect and measure comprise the terms on the technologies applied for monitoring of physiological parameters, such as “sensor”, “accelerometer”, “piezoelectric sensor”, “force sensing sensor”, “fiber optical”, “camera”, “radar”, “infrared sensors”, etc.

Method of measurement comprises the terms on the features of physiological parameters’ monitoring, such as “contactless”, “unobtrusive”, “smart home”, “wireless sensor network”, “nursing home”, etc.

The searches were carried out from November 2022 to January 2023. The search query is shown in Section A.1, Section A.2, Section A.3 and Section A.4 for all databases (IEEE Xplore, PubMed, Scopus, and Web of Science). In addition, this search was restricted to the inclusion/exclusion criteria and to the title and abstract of the paper for all databases.

A total of 6476 references (before removing duplicates) were collected from the databases (2298 from Web of Science, 1648 from Pub-Med, 1864 from Scopus, and 665 from IEEE Xplore). The records were downloaded in text format and, after removing duplicates, 3774 references were qualified for the data evaluation step. After evaluating the publications, 205 articles were selected by reading their titles and abstracts. Finally, the full text was read to ensure that the inclusion/exclusion criteria for the assessed articles were met. The number of studies included in the systematic review was 54. Statistical methods were used to analyse 44 publications.

### 2.3. Selection Process and Data Extraction

The studies identified by the search described in the previous section were imported into Citavi—a free reference management tool. Duplicates were then automatically identified and eliminated. The next step was to select the remaining publications, first by title and then by abstract. The titles and abstracts of non-duplicated publications were selected, and then all publications that were unrelated to the topic were removed.

The final step was the diagonal and full reading procedure of the publications selected in the previous stage, in order to extract information relevant to our research based on the inclusion/exclusion criteria. For this purpose, a predefined form was filled in with the data from the publication (manually, by a researcher). The fields in the form were as follows: paper name, list of authors, year of publication, information related to the stated research questions, measured physiological parameters related to cardiorespiratory activity, number of subjects participating in the studies, and type of device. In addition, the performance metrics for device evaluation were taken into account when considering papers—accuracy (including mean average or percentage error), sensitivity, and specificity. Those publications that consisted of reviews of multiple technologies, algorithms, or devices for cardiorespiratory activity monitoring were classified independently and reserved for answering the specific questions listed in the Introduction section.

### 2.4. Synthesis Methods

Statistical data visualisations were presented to aid in our understanding of the results. Statistics were performed using the assessment metrics and information on device characteristics for cardiorespiratory monitoring in the reviewed articles. Where possible, statistics were presented to explain the relevant information collected during the systematic review.

## 3. Results

### 3.1. Study Selection

Figure 1 shows the list of publications that met the inclusion/exclusion criteria and were selected for statistical analysis. The years of publication ranged from 2017 to 2022. The highest number of selected articles were published in 2022 (33.33% of all publications), and the lowest number were published in 2018 (3.70% of all publications). The research material selected for this systematic review was divided into two main groups: One group was called “Research and commercial systems and technologies” (see Figure 2), which included publications related to the use of devices (including prototypes) for cardiac and respiratory activity monitoring. Publications in this group were used for statistical analysis. The other group, known as “Other(s)”, included publications that provided information relevant to answering the research questions outlined in the Introduction, but that could not be included in the statistical analysis due to defined criteria.

Figure 2 shows the whole process that took place during this systematic review. It also shows the numbers of publications included and excluded at each stage of the review.

### 3.2. Study Characteristics and Individual Publications

Table 1 lists all publications in the group “Research and Commercial systems and technologies for cardiorespiratory monitoring”. This table provides information about the monitoring devices, such as the physiological parameters that could be measured and the numbers of subjects that participated in the experiments to evaluate the performance of the devices.

Table 2 shows all publications in the group “Research and commercial systems and technologies for cardiorespiratory monitoring”. This table also includes a summary of each publication.

### 3.3. Synthesis Results and Questions of Interest

The central and specific questions raised in the Introduction are answered and explained in detail in this section.

#### 3.3.1. MRQ: How Can the Cardiac Activity and Respiration Be Contactlessly Monitored during Sleep?

Based on the data presented in Table 1, five basic techniques can be identified for contactless and unobtrusive monitoring of cardiorespiratory activity. Non-contact monitoring can be performed by analysis of the following:Video images;Movements evoked by cardiac contractions (i.e., heartbeat movements);Subject movements (e.g., body, shoulders);Temperature maps and thermal images;Sounds of cardiac activity (e.g., heart sounds).

Some of the methods listed also have several directions without changing the overall approach to measurement (see Figure 3).

In the group of publications known as “Other(s)”, five studies deal with this question. Cardone et al. [67] stated that with thermal infrared imaging of the subject and image processing technologies, it is possible to extract heart rate, respiratory rate, and undoubtedly temperature (which is also a vital signal). Mendonça et al. [68] highlighted the use of respiratory analysis—alone or in combination with other sensors—as the method that gave the best results in detecting sleep apnoea and respiratory rate. In addition, Mendonça suggested that a combination of oximetry and sound analysis may be the best choice for respiratory analysis. Tran et al. [8] stated that with the increasing prevalence of OSA and its comorbidities—particularly cardiovascular comorbidities—there is a significant market potential for the realisation of non-contact continuous sleep monitoring technology. In addition, Tran provided several recommendations for improving research into the use of radar technology to monitor cardiorespiratory activity. At the same time, Sadek et al. [1] provided an overview of sensors using BCG technology (such as polyvinylidene fluoride film-based sensors, electromechanical films, strain gauges, hydraulic sensors, microbend fibre-optic sensors, fibre Bragg grating sensors, etc.) as a basis for contactless monitoring of HR and RR. Wang et al. [69] conducted a retrospective literature review and summarised the state of the art in using sensor technology for unobtrusive in-home health monitoring, including cardiac and respiratory activity. They divided sensors into 25 types that can be installed in rooms, electrical appliances, equipment, etc. These included motion sensors, contact sensors, pressure sensors, and current sensors.

Equally interesting is the statistical analysis of the data obtained. According to the information in Figure 4, the analysis of movements (in particular of the body, the shoulders, or the position of the subject) was observed in 38.46% of the studies. As this method is carried out using radar systems or radar technology, the statistical data confirm the trend in recent years towards the widespread use of this method for recording cardiorespiratory activity [23,25,28,30,35,38,39,43,47,50,52,55,57,60,61,62,64,70,71,72,73]. At the same time, the method of monitoring based on the analysis of heart sounds (9.62% of the total) also includes the use of radar in the hardware part of the system [26,31,37,49,58]. However, similar use of devices does not imply similar methods of obtaining the required information in these works, i.e., physiological parameters of cardiorespiratory activity.

As for the compliance with the trends of recent years, we cannot but mention the popularity of the method of analysing parameters of human vital signs using imaging technologies—e.g., smartphone cameras, depth cameras, infrared and RGB cameras [27,32,33,34,35,40,41,42,45,48,53,54,55,56,61,62,65]. Statistical data from our study support this position. Thus, 30.77% of the publications included in this review mentioned cameras as the central node of the measurement system. On the other hand, non-contact monitoring of cardiorespiratory activity based on the analysis of movements caused by cardiac contractions was mentioned in 13.46% of the publications. It is important to note that this method involves the use of measurement techniques such as BCG and seismocardiography (SCG), which are long-established trends in the recording of human vital signs [29,36,44,46,51,59,63]. The last (but not least) group of methods consisted of papers that used the analysis of temperature and temperature maps (images) of the subject as a non-contact monitoring method—7.69% [27,48,56,65].

Finally, it is important to note that some publications used several of these analysis methods simultaneously—in particular, the analysis of video images and the subject’s movements using cameras and radar, respectively [61,62]. For this reason, these papers were counted twice in the statistical analysis. At the same time, it is possible to detect the use of different technologies of the same method—for example, the use of two different types of cameras (e.g., infrared and RGB)—for the same analysis method [27,35,48,56,66]. However, the publication data were counted once for statistical analysis.

#### 3.3.2. SRQ-1: Which Technologies Can Be Used for the Contactless Measurement of Cardiac Activity and Respiration during Sleep?

In this subsection, we would like to focus the reader’s attention directly on an overview of technologies that can be used for contactless monitoring of cardiorespiratory activity during sleep. Of course, a description of the technology implies a description of the sensors used within a particular technology. However, the sensors are described in the following subsection.

As shown in Figure 3 and Figure 4, there are several methods for contactless and unobtrusive monitoring of cardiorespiratory activity during sleep. In addition, it has already been noted that some of the methods mentioned involve the use of similar technologies for monitoring. In particular, we are talking about the use of radiolocation or interferometry technologies implemented by radar applications. These can be used to extract HR and RR by analysing the chest, shoulder girdle, and body movements or sound waves from heart activity. The same physiological parameters of vital signs and temperature can be analysed from video or temperature images obtained using colour and infrared cameras. This technology is called remote or photoplethysmography (rPPG or iPPG). At the same time, heatmaps and image analysis also refer to technologies such as thermography. In addition, the increasing popularity of BCG or motion analysis of cardiac contractions has already been noted [7,29,36,44,46,51,59,63]. This method involves the use of a wide range of sensors, which will be discussed in the following section. Finally, an equally interesting application is the infrared thermal pattern analysis technology (IR technology) [24]. Table 3 provides information on the above technologies used in the publications included in this review. It is important to note that publications using more than one technology have been counted twice [27,42,62,64,65].

Below is a brief description of each of the above technologies for a fuller and more accurate understanding of their operation.

Interferometry

Interferometry is a technique that uses the interference of superimposed waves to extract information [74]. Interferometry is widely used in science and industry to measure small displacements, refractive index changes, and surface irregularities. Interferometry uses the principle of superposition to combine waves in such a way that the result of their combination has a meaningful property that is diagnostic of the original state of the waves.

In the context of contactless measurement of physiological parameters of vital signals using the interferometric method, the best-known device is radar. In most of the work carried out with radar, a voltage-controlled oscillator is used to generate an RF signal that is sent through a transmitting antenna. In turn, the electromagnetic wave (signal) transmitted by the system is reflected from the subject’s chest and then picked up by the receiving antenna [26,52]. It is important to note that depending on the type and design of the radar, the antennas may be combined. In the next stage, the received signal is filtered, amplified, and digitised to extract the necessary information about the signal. This method is characterised by a fairly high accuracy in the measurement of heart rate and respiratory rate, although the accuracy depends on the distance and position of the radar relative to the patient. This aspect may affect the unobtrusiveness of the system to the patient. However, it is worth noting that none of the selected articles highlighted the appearance of discomfort in the subjects. In addition to that, in some of the works mentioned in this review, studies were carried out to investigate the dependence of the results on the distance between the patient and the radar [30,37,41,43,52,57,58,64].

Ballistocardiography (BCG)

Ballistocardiography (BCG) is a non-invasive method of producing a graphical representation of the repetitive movements of the human body caused by the heartbeat. These repetitive movements are caused by the rapid acceleration of blood as it is ejected and travels through the major vessels of the body during periods of relaxation and contraction—known as diastole and systole, respectively [75]. During atrial systole, as blood is ejected into the great vessels, the centre of mass of the body shifts towards the head of the body. This shift in the centre of mass of the body produces the BCG waveform as the distribution of blood changes during the cardiac cycle [76].

BCG is currently generating a lot of interest due to the information technology revolution, including hardware technology, software, and services. We can embed BCG sensors in the environment without the need for medical personnel [1]. In terms of the variety of sensors used for this method, the most popular sensors are accelerometers [13], gyroscopes [77], and inertial measurement units. The variety of sensors for this technique is presented in detail in the following subsection. It is also impossible not to mention the variety of BCG signal processing algorithms, an overview of which can be found in the work of Sadek et al. [1].

Remote photoplethysmography (rPPG)

Photoplethysmography (PPG) is a method of measuring the parameters of vital physiological signals [78]. PPG measures changes in tissue’s light absorption due to the pulsatile nature of the cardiovascular system and changes in blood volume [79]. Due to the potential disadvantages of contact PPG, non-contact methods of recording the PPG signal have been demonstrated [80]. In fact, rPPG is the non-contact equivalent of the reflective mode of PPG, using ambient light as the source and the camera as the receiver. The light reflected from the skin is then evaluated by capturing the subtle changes in skin colour with the camera as the blood volume changes. Several image and signal processing steps produce a pulsed signal, also called the rPPG signal.

Several biomedical parameters can potentially be measured with rPPG or PPG signals, including heart rate, pulse rate variability, respiratory rate, vascular occlusion, peripheral vasomotor activity, and blood pressure by transit time [81].

Typically, an rPPG signal evaluated by any of the possible analysis methods is noisy, due to the estimation method, lighting variations, internal digital camera noise, and motion [78]. Similar to BCG, there are many processing methods for the rPPG signal [82,83,84].

Thermography and Infrared (IR) technologies

In a sense, thermography (including infrared) can be classified as an rPPG method in conjunction with the use of cameras to measure both techniques. At the same time, infrared thermography (IRT), also known as thermal imaging, is a remote, non-contact, and passive monitoring approach that detects the radiation naturally emitted by an object (e.g., human skin) [85]. However, the main advantage of thermal imaging over other imaging techniques is that it does not require a radiation source [86,87]. It is also important to note that this method can be used in low-light conditions [88]. Facial IRT can provide insight into an individual’s autonomic activity by assessing temperature changes over time and spatial patterns [89].

In one way or another, this method is based on analysing the patterns of the obtained image of a person’s face or body in the thermal (infrared) range of radiation. Recently, machine learning (ML)-based approaches have been used to analyse IRT data to improve the technology’s ability to assess pathologies and to enhance emotion recognition in human–machine interaction [90,91].

#### 3.3.3. SRQ-2: Which Sensors Are Used for Those Technologies?

Based on the data presented earlier (Figure 4 and Table 3) on the methods and technologies used, it is also possible to answer the question of which sensors are used for these technologies. The sensors used in the papers included in this review can be divided into four groups: temperature sensors, motion sensors, radars, and cameras. A more detailed breakdown of the sensors is shown in Figure 5.

As can be seen, the pyroelectric infrared sensor was applicable among temperature sensors to determine physiological parameters that can be used to monitor the subject’s chest movements and analyse the resting heart rate (RHR). In addition, changes in the infrared thermal spectrum can be detected to assess this parameter [24]. For thermography and remote plethysmography technologies, cameras are used as the hardware. The cameras can be divided into RGB [27,32,33,42,45,48,53,54,56,61,62,65], infrared (IR) [27,40,48,56,65], and depth cameras [34,35,41], based on the data obtained. We also distinguish smartphone cameras as a separate group, as among the selected publications we can find several papers where the study was carried out using this particular technology [53,54,66]. At the same time, BCG and SCG technologies often use motion sensors such as accelerometers [51], seismographic sensors [36], tensometric sensors [44], piezoelectric sensors [59], vibroacoustic sensors [63], sensors on polyvinylidinochloride films [29], and fibre-optic sensors [46]. Finally, using interferometry technology, devices such as radars are applicable for monitoring cardiorespiratory activity. Among them, we can distinguish Doppler [23,26,31,50], ultrasonic [30], ultra-wideband [25,28,35,62,64,72], and continuous radars [26,31,37,38,39,43,49,52,55,57,58,60,61]. Table 4 shows the frequency range used by these radars and the number of publications included in this review where the relevant radar type was mentioned. Continuous-wave (CW) radar is the most popular for monitoring cardiac and respiratory activity.

#### 3.3.4. SRQ-3: Which Physiological Parameters Can Be Extracted from Those Sensors?

According to the data presented in Table 1, and using the types of sensors mentioned above, it is possible to extract physiological signal parameters such as the following:Heart rate (HR);Respiration rate (RR);Temperature;Blood pressure.

It is worth noting that these are all basic parameters of human vital signals. In addition, it is possible to extract complementary parameters such as heart rate variability (HRV) [47,48,49,59,63] and breathing rate variability (BRV) [59], as well as awakening epoch [29] and sleep apnoea [23,30,39,62,65]. However, we focused on the main physiological parameters in the statistical analyses. Figure 6 and Figure 7 present information on the number of parameters recorded for the systems and the frequency of recording of parameters in the publications included in the review, respectively. As can be seen in Figure 6, contactless monitoring of one of the physiological parameters was predominantly performed in 66.67% of the papers. The registration of two parameters at the same time was found in a much smaller number of papers—27.08%. Only three papers presented systems that recorded three physiological parameters at the same time—6.25%. Negishi et al. [27] presented a system for measuring a patient’s heart rate, respiratory rate, and facial temperature by evaluating IR and RGB heatmaps. By measuring the blood volume pulse (BVP) using an RGB camera, Negishi et al. [42] were able to measure the physiological parameters of the heart and respiratory signals and body temperature. In addition, Talukdar et al. [66] used a smartphone camera and a dedicated app to estimate HR, BF, and blood pressure. Thus, according to the data in Figure 7, it is impossible to clearly define which parameter is registered more often, as HR and RR are marked in an equal quantity of publications. However, when two or three physiological values are recorded at the same time, HR and RR are recorded as a pair.

#### 3.3.5. SRQ-4: What Are the Medical Applications of Contactless Cardiac and Respiratory Monitoring during Sleep?

The devices and systems for contactless and unobtrusive monitoring of cardiorespiratory activity considered in this review can be categorised according to their medical applications and their focus on issues and problems, as follows:Cardiac activity;Respiratory activity;Sleep medicine.

At the same time, it is essential to note that each of the above groups of medical applications includes more specialised issues. More detailed information is provided in Figure 8. According to this figure, devices related to cardiac issues and ubiquitous and routine activity monitoring address more specific issues. These include the detection of heart failure [50] and arrhythmias [25,47], monitoring of heart rate variability (as noted earlier), and prevention of haemorrhages [50].

The medical applications of respiratory activity also address issues in addition to monitoring. These include the prediction and prevention of influenza [27,42] and monitoring of respiration in coronavirus infection (COVID-19) [60]. In addition, the medical applications of the devices included in the review include the detection of Cheyne–Stokes respiration [23,31] and biota [65].

The medical applications related to sleep medicine include the detection of sleep apnoea [30,39,62,65] and sleep monitoring [29,30].

A detailed breakdown of the medical applications for cardiorespiratory monitoring technologies is shown in Figure 8. The statistical data (see Figure 9) correspond to the information on the different medical applications of contactless monitoring of cardiorespiratory activity.

#### 3.3.6. SRQ-5: What Are the Differences in the Quality of the Measurements (Contactless vs. Contact-Based/Attached Devices)?

Probably one of the most interesting and challenging specific issues for our review is the question of the differences in measurement quality between contact and non-contact monitoring. In other words, it is necessary to clearly define the capabilities of each of the systems included in this study, their accuracy, and their characteristics compared to recognised standard monitoring methods, which were taken as the reference systems and devices. In our work, we focused on evaluation metrics—i.e., the characteristics of non-contact monitoring systems. However, we noted the most common systems (i.e., standards) used in the publications included in this review, against which the comparisons were made. For example, systems such as electrocardiographs [25,72,73], patient monitors [26,49,73], pulse oximeters [33,56], Holter monitors [63], polysomnography (PSG) systems [23,29,31], PPG sensors [24,59], BioPAC systems [50,61] and the Hexoskin commercial system [71] were frequently used for cardiac reference signal extraction. Moreover, respiratory belts [54,55], spirometers [34], Plux systems [45], impedance pneumographs [28,40], capnography systems [41], and even manual counting [52] have been used to obtain respiratory signals. In addition, Zephyr BioModule [53] has been used for the parallel registration of cardiac and respiratory signals. Digital thermometers and sphygmomanometers have been used to register temperature and blood pressure [42,70].

To answer this question, we had to divide non-invasive monitoring systems and technologies according to the physiological parameter being monitored. In our case, the division was made into systems evaluating heart rate and systems evaluating respiratory rate. This was explained by the existence of works where the systems evaluated both parameters and the metrics used to evaluate the physiological parameters were different. However, after this separation, it was found that no universal and unified metric could be identified to compare the systems. The following metrics were identified in the publications included in our review:Accuracy, sensitivity, specificity (Acc, Sen, Spe);Mean absolute error (MAE);Mean absolute percentage error (MAPE);Pearson’s correlation coefficient (r-Pearson).

As it is impossible to compare the listed metrics with one another, and comparing systems within each metric does not allow us to clearly identify the system with the best parameters, we decided not to separate them further. Table 5 provides information on the estimated metrics of the systems considered for respiratory rate estimation. Table 6 provides similar information for the heart rate estimation systems. In some cases, results are presented for multiple metrics simultaneously, allowing for a better assessment of the capabilities of non-contact monitoring systems and technologies.

#### 3.3.7. Enhancements, Advantages, and Limitations

In this subsection, we would like to address the potential limitations of each of the non-invasive cardiorespiratory monitoring technologies discussed in this review. It is worth noting that this aspect was partially covered in the technology description in Section 3.3.2, but not all technologies were given sufficient attention.

When discussing the use of interferometry (i.e., the use of radar in HR and RR estimation), we have already mentioned the relatively high accuracy of the results. However, the accuracy and quality of the results obtained using this method are highly dependent on the distance between the device and the patient, as well as on the presence of possible obstacles in the path of the electromagnetic wave. It should also be noted that the radar is not easy to maintain and, in the event of a failure, it is difficult to replace the unit/module at home for the user.

As noted in Section 3.3.3, ballistocardiography has a fairly wide range of sensors for recording cardiorespiratory activity. Therefore, the limitations of this technique also vary. For example, the use of fibre-optic or polyvinylidene chloride film sensors allows HR and RR to be assessed with high accuracy (minimal mean data error). However, the cost of this technology, combined with the expense and inability of the user to replace the fibre or other modules in the system themselves, limits its use in the home. Systems based on piezoelectric sensors are no less accurate. However, one of the main limitations of piezoelectric sensors is that they are sensitive to vibration or acceleration, which is common when the system is placed under a bed mattress. In other words, further thought needs to be given to the possible positioning of the system and sensors to minimise the disadvantages of the system. As mentioned earlier, accelerometers (as well as seismographic sensors) have become the most popular sensors for BCG analysis of cardiorespiratory activity. The ability to measure in all directions of the patient’s chest expansion has allowed scientists to obtain the most comprehensive picture and parameter readings of physiological signals. At the same time, we would like to point out that the number of studies using this type of sensor is low. This is due to the fact that, when selecting the articles, we often found that the accelerometer was in contact with the patient (on the chest), which automatically excluded such works from consideration. In turn, the accuracy of the data obtained with this type of sensor is affected not only by the location of the system [92], but also by the signal processing algorithm. At the same time, such systems or devices are characterised by their ease of maintenance and replacement of modules in the event of failure at home, which is an undeniable advantage over other technological solutions.

When considering the limitations of remote photoplethysmography, it is worth highlighting the potential for camera failure and the potential intrusiveness of the light source to the patient. In other words, when using this technology, the system should be installed in a way that causes the least discomfort to the patient. At the same time, rPPG is also highly accurate, thanks to the use of machine learning and neural network techniques in image analysis and the estimation of cardiorespiratory parameters. On the other hand, in the case of infrared thermography—which, as mentioned above, can be used in cases where there is no acceptable light source for the camera—there are two possible limitations to the use of the technology: The first is the installation of the system to ensure unobtrusiveness to the patient. The second is the potential problems with infrared data acquisition if the area of interest to the system (e.g., the patient’s face or chest) is not fully covered by the camera lens or is under a blanket. However, the accuracy of the data obtained is high, again due to the use of machine learning or neural network techniques in image analysis. Summarising the limitations, advantages, and disadvantages of the technologies considered, it can be said that each of the methods has a fairly high accuracy in estimating cardiorespiratory activity parameters. At the same time, the possible limitations of these methods are mainly in the technical component.

In terms of potential limitations, it should be noted that it may be difficult to implement the non-contact cardiorespiratory monitoring technologies listed in this review in a realistic clinical or home setting. Of the papers listed, only a small number of systems have been tested directly in hospital wards or at home. At the same time, the authors claim that the developed systems can be used at home or in hospital. However, when it comes to systems that need to be placed under a bed or mattress, all of the nuances of sensor and system installation need to be considered, depending on the bed model and possible room layout (e.g., compact room dimensions or the distance of the bed from the power supply). In addition, although obvious, detailed training is required before the system is installed for the patient at home, and regular opportunities must be provided for advice on operation or action in the event of questions or malfunction of the system. In the clinical setting, the installation of non-contact cardiorespiratory monitoring systems should include introductory training for nursing staff in the initial maintenance of the systems. This, in turn, may be a problem in allocating workload to nursing staff in the early stages. In any case, wherever the system is to be used, end-user training and the production of a user manual with detailed explanations of how to operate the system and ensure accurate and reliable measurement of cardiorespiratory parameters are required.

As for the possible future directions of technological development, we can mention the emergence and development of complementary technological solutions, as well as the improvement of processing algorithms in order to expand the system’s capabilities. Thus, we can give an example of the successful validation of the system based on linear resistive pressure sensors for the evaluation of cardiorespiratory activity of [93]. We believe that this example, along with possible alternative technological solutions, is an attractive direction for researchers.

## 4. Discussion

In order to provide a sample of the relevant literature, this paper conducted a limited search of four databases—namely, Web of Science, IEEE Xplore, PubMed, and Scopus—to identify works in the field of contactless and unobtrusive cardiorespiratory monitoring during sleep. This search strategy may not be able to provide exhaustive and comprehensive coverage of the literature; however, we believe that the sampled literature adequately reflects the current state of research in the field of unobtrusive health monitoring. This study had a high enough sensitivity of this retrieval strategy—1.4% of the initially returned papers were included for in-depth textual analysis (53/3774 = 1.40%). Based on the terminology, the included literature was reviewed in a structured way. Returning to the initial questions proposed at the beginning (Section 1), we continue to answer them.

When we analysed the publications included in this review to answer the main research question (how can cardiac activity and respiration be contactless monitored during sleep?), we found that the monitoring of cardiorespiratory activity by motion analysis is mainly performed by receiving and transmitting radar signals in the direction of the region of interest. Such an area could be the patient’s chest or shoulder girdle. In addition, the patient’s sleeping position can also serve as a region of interest. It should be noted, however, that the method of monitoring by analysing heart sounds involves the same techniques, but the region of interest is the sound waves recorded from the region of interest. Image analysis can also be used to monitor cardiorespiratory activity. This is mainly possible by recording the patient’s face and upper body. Blood circulation in the body, changes in thoracic and/or abdominal depth, changes in landmarks, and changes in skin colour during respiration are the main trends of physiological parameter estimation in image analysis. At the same time, cardiorespiratory monitoring is also performed using temperature analysis, which is possible by looking at heatmaps and patient images. When implementing monitoring using heartbeat-induced motion recording, it is important to note the wide variety of measurement approaches. The most popular approach is to capture the oscillations resulting from the ventricular contraction of the heart and the subsequent ejection of accelerated blood into the aorta. Equally attractive are approaches based on measuring the electrical signal derived from the recoil forces in the body due to blood flow, or measuring the subject’s body pressure.

Based on the literature review, we can identify five main technologies for contactless cardiorespiratory monitoring during sleep—interferometry, thermography, remote photoplethysmography, ballistocardiography, and infrared technology. These technologies are related to the question of their application to the non-contact measurement of cardiac activity and respiration during sleep (SRQ-1).

We grouped the sensors and systems for contactless monitoring into four types: temperature sensors, radar systems, motion sensors, and camera-based systems. The peculiarities of the sensors and systems in each group are presented while answering the question on the use of sensors for the abovementioned technologies (SRQ-2).

The main physiological parameters extracted during this research in cardiorespiratory monitoring are HR and RR (SRQ-3). As an additional result, it is possible to calculate HRV and BRV, which are also important in clinical practice. It is also possible to assess temperature and blood pressure, known as human vital signs (with HR and RR), using the systems and technologies under consideration. Finally, it is important to note that sensor applications allow us to determine wake times and estimate sleep efficiency.

Looking at the statistical data on the various medical applications of contactless monitoring of cardiorespiratory activity (SRQ-4), it can be seen that cardiac and respiratory issues are covered to about the same extent, with a slight preponderance in favour of respiration. This can be explained by the fact that the respiratory signal usually requires fewer instruments and fewer operations for signal processing. In addition, these operations are often simpler in their description, structure, and implementation.

Sleep medicine issues have been addressed to a much lesser extent. It is important to note that the issues covered for this medical application also relate to respiratory and cardiac issues but have been placed in a separate group. The reason for this is that in order to obtain and analyse the information, it is necessary to carry out a long-term experiment in which the patient has to stay awake for some time. At the same time, the issue of sleep monitoring requires the tracking of physiological parameters of both respiratory and cardiac signals [39]. However, Park et al. [29] only performed data analysis (including the subject’s awakening) on the respiratory signal in their paper.

Based on the data on the performance of respiratory activity monitoring systems, several studies stand out for each of the previously outlined metrics (SRQ-5). For metrics such as accuracy, sensitivity, and specificity, a relatively high accuracy (94%) combined with an acceptable mean absolute percentage error (6.25%) stands out in the work of Ullal et al. [51]. This study seems to be the preferred one, despite the higher accuracy values in the work of Han et al. [64] (98% and a mean absolute error of 0.23 bpm), Do et al. [55] (98%), and Shokouhmand et al. (98%) [61]. The reason for this is that, despite their high scores and study perspectives (an additional factor in the analysis), the abovementioned studies had fewer subjects participating in the system validation. Several papers can be distinguished when analysing systems in terms of mean absolute error values. For example, Khushaba et al. [23] obtained an error value of 0.38 bpm by sending and receiving a reflected wave from a sensor and recording body motion with radar; Wang et al. [52] obtained a value of 0.49 bpm by measuring the phase delay of the reflected radio signal; He et al. [62] obtained an error of 0.61 bpm using radiolocation and remote plethysmography techniques. At the same time, when comparing works on the mean absolute percentage error value, the work of Mateu-Mateus et al. [45] stands out, who achieved an error value of 6% by calculating the changes in the position of the detected image pattern over time. We should also note the works of Villaroel et al. [33], Rossol et al. [40], and Gwak et al. [54], who obtained relatively high values for the Pearson correlation coefficient: 0.980, 0.948, and 0.934, respectively. However, the highest value of this parameter (0.992) was obtained by Xu et al. [59], but it is important to note that their study involved six subjects.

The performance of cardiac monitoring systems (i.e., HR recording) was assessed in the 27 papers included in this review. Several papers stand out for the high accuracy values achieved by the monitoring systems. The work of Ullal et al. [51], as in the case of respiratory monitoring, had an accuracy of 97% and a percentage error of 3.6% in determining HR. Similar accuracy was shown by the works of Park et al. [36], Parciani et al. [63], and Shi et al. [49], using different techniques—SCG, BCG, and interferometry, respectively. Yu et al. [48] achieved 99% accuracy in HR estimation by measuring temperature maps and images of subjects’ faces using infrared and conventional cameras. Kapu et al. [24] and Xu et al. [43] achieved 95% accuracy for the estimation of a physiological parameter. Comparing the systems in terms of MAE, quite reliable results were shown by Talukdar et al. [66]—2.9 bpm with over 400 subjects. At the same time, Chen et al. [46] and Ling et al. [57] estimated HR with errors of 1.17 and 1.28 bpm, respectively. However, their studies involved much smaller numbers of subjects, so it is not possible to compare all three papers.

## 5. Conclusions

There is a large body of literature on non-invasive cardiorespiratory monitoring. However, there is a lack of information on methods and techniques that have been validated and can be unobtrusively introduced into clinical practice. In this article, we conducted a systematic review of publications related to these technologies. We have analysed and briefly described non-contact cardiorespiratory monitoring techniques and technologies, along with their potential medical applications. The most popular methods are interferometry (using radar), remote photoplethysmography (using RGB cameras), infrared thermography, and ballistocardiography, which allows a wide range of uses. We also compared their performance with several existing, recognised contact systems that are commonly used as standard techniques in medicine. In addition, we have speculated on possible future directions for the development of signal processing technologies and algorithms to assess the physiological parameters of vital signs. Furthermore, we suggested possible limitations and difficulties in implementing the technology in real clinical and home settings. The obtained data on the methods and technologies of non-contact and unobtrusive monitoring of cardiorespiratory activity during sleep suggest current trends and directions of the development of medical non-contact, non-invasive, and unobtrusive systems and technologies related to the monitoring of physiological parameters of vital signals in sleep medicine. In addition, this review might be helpful in actualising the current state of the art in the relevant research topic.

## Figures and Tables

**Figure 1 sensors-23-05038-f001:**
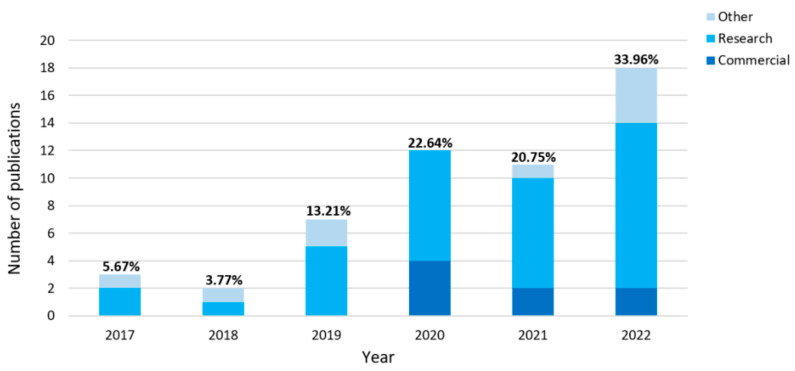
Bar plots with the numbers of publications that met the inclusion/exclusion criteria. The terms “Commercial” and “Research” refer to the purpose of cardiorespiratory monitoring. “Other” refers to the other publications selected for this systematic review (e.g., systematic reviews).

**Figure 2 sensors-23-05038-f002:**
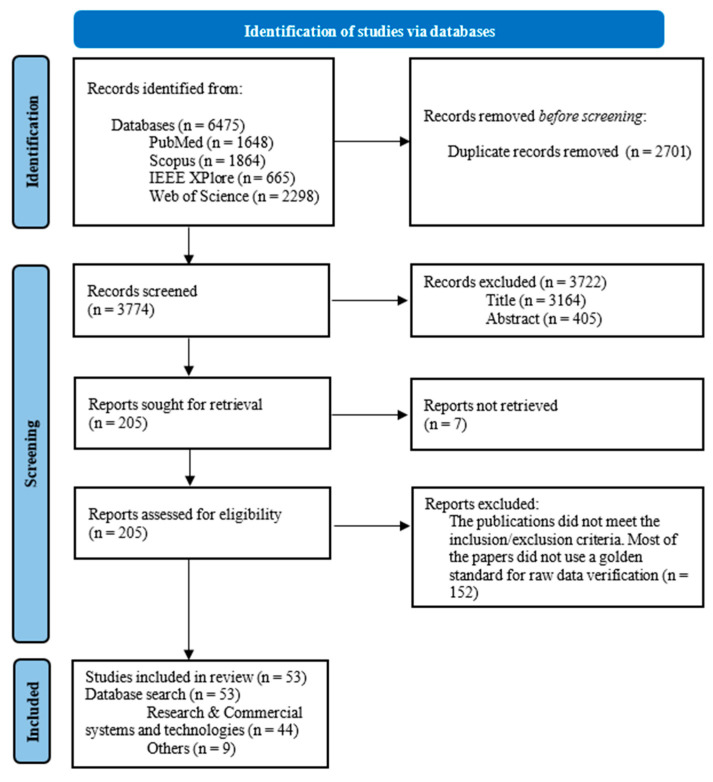
PRISMA 2020 flowchart for selecting the entire set of the included publications.

**Figure 3 sensors-23-05038-f003:**
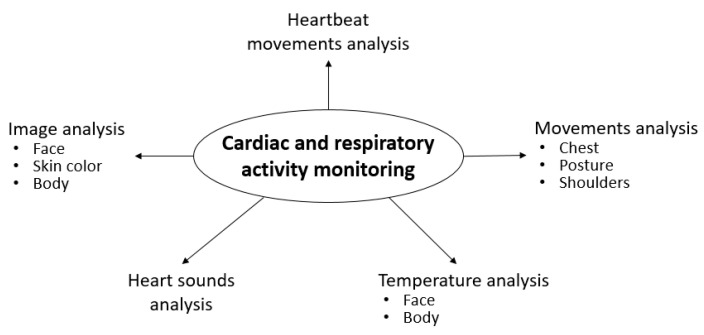
Techniques for contactless monitoring of cardiac and respiratory activity.

**Figure 4 sensors-23-05038-f004:**
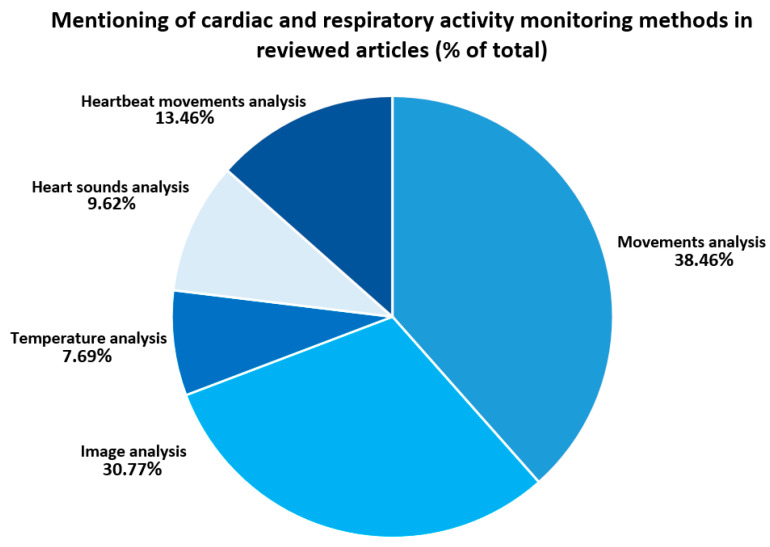
Papers’s percentage usage of considering cardiorespiratory monitoring methods.

**Figure 5 sensors-23-05038-f005:**
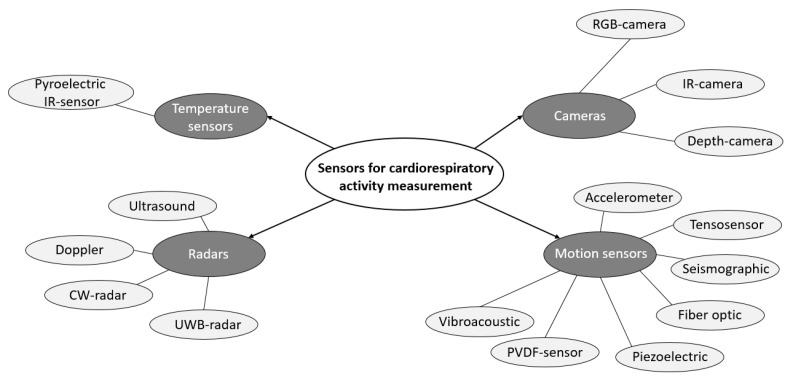
A detailed breakdown of the applicable sensors for cardiorespiratory activity.

**Figure 6 sensors-23-05038-f006:**
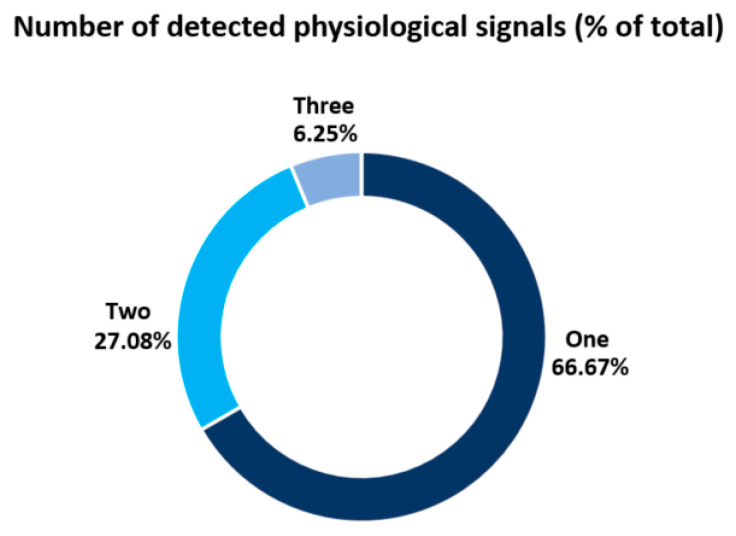
Number of detected physiological parameters and their percentages (of total papers).

**Figure 7 sensors-23-05038-f007:**
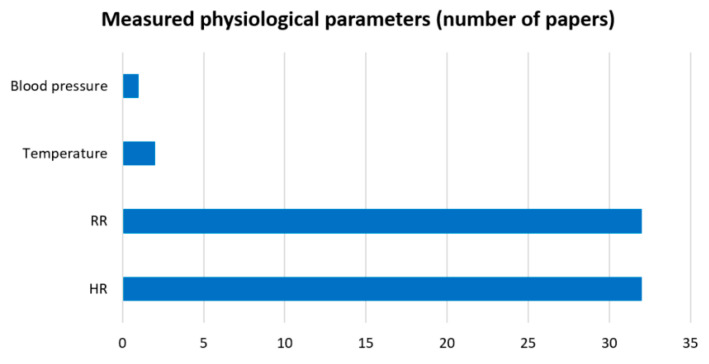
Frequency of registration parameters among the included publications.

**Figure 8 sensors-23-05038-f008:**
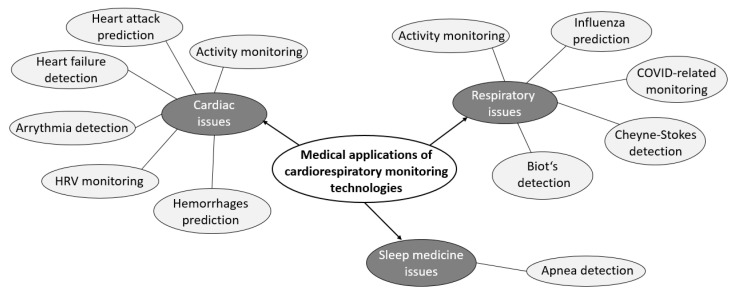
A detailed breakdown of the medical applications of cardiorespiratory monitoring technologies.

**Figure 9 sensors-23-05038-f009:**
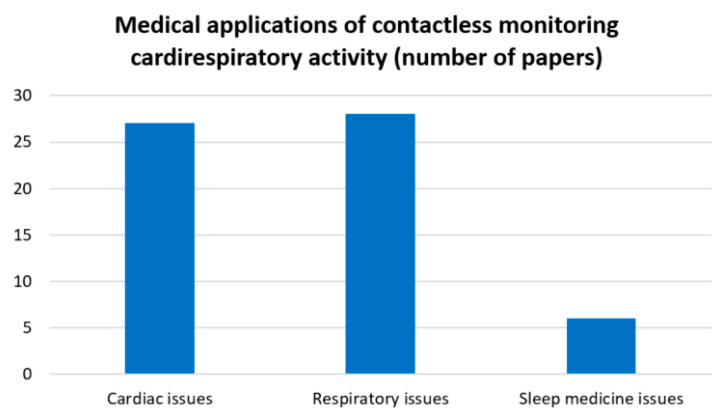
Diagram of medical applications’ amounts.

**Table 1 sensors-23-05038-t001:** Publications (Research and Commercial devices for cardiorespiratory monitoring) that met the inclusion/exclusion criteria. HR—heart rate, RR—respiratory rate.

Publication, Year	Monitoring Type	Measured Physiological Parameters	Number of Subjects	Type of Device
[23], 2017	Body movement analysis	RR	52	Research
[24], 2017	Chest movement analysis	HR	30	Research
[25], 2018	Chest movement analysis	HR	22	Research
[26], 2019	Heart sounds analysis	HR, RR	30	Research
[27], 2019	Face image analysis	HR, RR, temperature (facial)	25	Research
[28], 2019	Chest movement analysis	RR	42	Research
[29], 2019	Body movement analysis	RR	30	Research
[30], 2019	Chest movement analysis	RR	12	Research
[31], 2020	Heart sounds analysis	HR, RR	10	Research
[32], 2020	Face image analysis	HR	5	Research
[33], 2020	Face image analysis	HR, RR	40	Commercial
[34], 2020	Body image analysis	RR	39	Commercial
[35], 2020	Body movement analysis	RR	5	Research
[36], 2020	Heartbeat movement analysis	HR	41	Commercial
[37], 2020	Heart sounds analysis	HR, RR	11	Research
[38], 2020	Chest movement analysis	RR	16	Research
[39], 2020	Chest movement analysis	HR, RR	30	Research
[40], 2020	Body image analysis	RR	17	Research
[41], 2020	Body image analysis	RR	14	Commercial
[42], 2020	Body and face images analysis	HR, RR, temperature (body)	50	Research
[43], 2021	Chest movement analysis	HR	40	Commercial
[44], 2021	Heartbeat movement analysis	HR	20	Research
[45], 2021	Body image analysis	RR	21	Research
[46], 2021	Heartbeat movement analysis	HR, RR	11	Research
[47], 2021	Body movement analysis	HR	6	Research
[48], 2021	Face image analysis	HR	20	Research
[49], 2021	Heart sounds analysis	HR	25	Research
[50], 2021	Body movement analysis	HR	22	Research
[51], 2021	Heartbeat movement analysis	HR, RR	45	Commercial
[52], 2021	Body movement analysis	HR, RR	12	Research
[53], 2022	Body image analysis	HR, RR	18	Research
[54], 2022	Body and face images analysis	RR	30	Research
[55], 2022	Chest movement analysis	RR	32	Commercial
[56], 2022	Skin colour image analysis	HR	42	Research
[57], 2022	Body movement analysis	HR	20	Research
[58], 2022	Chest movement analysis	HR	15	Research
[59], 2022	Heartbeat movement analysis	HR, RR	6	Research
[60], 2022	Chest movement analysis	RR	30	Research
[61], 2022	Postures and chest movement analysis	HR, RR	10	Research
[62], 2022	Body movement and images analysis	RR	17	Research
[63], 2022	Heartbeat movement analysis	HR	24	Research
[64], 2022	Body movement analysis	RR	15	Research
[65], 2022	Face image analysis	RR	10	Research
[66], 2022	Face image analysis	HR, RR, blood pressure	463	Commercial

**Table 2 sensors-23-05038-t002:** Publications (Other(s)) that met the inclusion/exclusion criteria.

Publication, Year	Objective	Type of Publication
[67], 2017	Review of the principal achievements of thermal infrared imaging in computational psychophysiology, focusing on the capability of the technique for providing ubiquitous and unwired monitoring of psychophysiological activity and affective states	Review article
[68], 2018	Reviewing publications that show the performances of different devices for the ambulatory diagnosis of sleep apnoea	Review article
[8], 2019	A comprehensive review of the current state of non-contact Doppler radar sleep monitoring technology, providing an outline of current challenges and making recommendations on future research directions to practically realise and commercialise the technology for everyday usage	Review article
[1], 2019	Review of the sensors used for obtaining BCG signals. Review of the signal processing methods as applied to the various sensors to analyse the BCG signal and extract physiological parameters such as heart rate and breathing rate, as well as determining sleep stages	Review article
[69], 2021	A retrospective literature review and summarised the state-of-the-art research on leveraging sensor technology for unobtrusive in-home health monitoring	Review article
[70], 2022	Performance of validation results on the use of Lifelight software to accurately cardiorespiratory signal measurement	Commercial article
[71], 2022	A complete framework for vital sign processing using a 77 GHz FMCW radar	Research article
[72], 2022	Development of two complementary heartbeat signal restoration methods to perfectly recover heartbeat signal variation based on the analysis of the properties of UWB signals containing heartbeats and respiration	Research article
[73], 2022	A contactless and multiscale cardiac motion detection method is proposed, with no blind detection of segments during the entire cardiac cycle	Research article

**Table 3 sensors-23-05038-t003:** Technologies for monitoring cardiac and respiratory activity. RGB—red, green, and blue colour; IR—infrared.

Technology	Amount of Papers
Interferometry	21
BCG (incl. SCG)	7
rPPG (incl. iPPG)	16
Thermography	4
IR technologies	1

**Table 4 sensors-23-05038-t004:** Radars for monitoring cardiorespiratory activity. UWB—ultra-wideband, CW—continuous wave.

Radar Type	Frequency Range, GHz	Number of Papers
Ultrasound	4∙10–5	1
Doppler	2.4 … 10.525 24	3
UWB radar	2.9 … 10.1	6
CW radar	24 60 77	15

**Table 5 sensors-23-05038-t005:** Comparison of the estimated technologies’ characteristics (for respiratory activity measurement). Acc—accuracy, Sen—sensitivity, Spe—specificity, MAE—mean absolute error, MAPE—mean absolute percentage error, Sub—number of subjects.

Publication, Year	Evaluation Metrics						Additional Points	Sub	Publication Type
	Acc	Sen	Spe	MAE, bpm	MAPE, %	r-Pearson			
[23], 2017	86	71	88	0.38	-	-		52	Research
[26], 2019	93	93	-	-	-	0.914		30	Research
[29], 2019	76	-	-	-	-	-	<1 bpm	30	Research
	97						<2 bpm		
	99						<3 bpm		
[28], 2019	-	-	-	-	-	0.893	Depends on movement level (from lack to maximum)	42	Research
						0.833			
						0.749			
[30], 2019	-	-	-	-	-	0.977	0.5 m distance	12	Research
						0.956	1 m distance		
						0.844	2 m distance		
						0.648	3 m distance		
[27], 2019	-	-	-	-	-	0.920		25	Research
[42], 2020	-	85	90	-	-	0.87		50	Research
[35], 2020	-	-	-	1.52	-	-	For single subject	5	Research
				1.32			For multiple subjects		
[38], 2020	-	-	-	1.5	-	0.870		16	Research
[33], 2020	-	-	-	2.8	-	0.980		40	Commercial
[34], 2020	-	-	-	-	10.7	-	With T-shirt	39	Commercial
					14		Undressed		
[31], 2020	-	-	-	-	9.1	0.910		10	Research
[40], 2020	-	-	-	-	-	0.948		17	Research
[41], 2020	-	-	-	-	-	0.910		14	Commercial
[51], 2021	94	-	-	-	6.25	-		45	Commercial
[52], 2021	-	-	-	0.49	-	-		12	Research
[46], 2021	-	-	-	2.16	-	-		11	Research
[45], 2021	-	-	-	-	6	0.850		21	Research
[60], 2022	80	-	-	-	-	-	6 bpm accuracy	30	Research
	97						10 bpm accuracy		
[61], 2022	98	-	-	-	-	-		10	Research
[62], 2022	90	-	-	0.61	-	-	For 1 subject	17	Research
				0.68			For 2 subjects		
[64], 2022	95	-	-	0.18	-	-	Not in real time	15	Research
	98			0.23			Real time		
[55], 2022	98	-	-	-	-	-		32	Commercial
[65], 2022	95	-	-	-	-	-		10	Research
[54], 2022	-	-	-	1.95	-	0.886	For head	30	Research
						0.934	For chest		
[53], 2022	-	-	-	1	-	-		18	Research
[66], 2022	-	-	-	2.9	-	-		463	Commercial
[59], 2022	-	-	-	-	-	0.992		6	Research

**Table 6 sensors-23-05038-t006:** Comparison of the estimated technologies’ characteristics (for cardiac activity measurement). Acc—accuracy, Sen—sensitivity, Spe—specificity, MAE—mean average error, MAPE—mean average percentage error, Sub—number of subjects.

Publication, Year	Evaluation metrics						Additional Points	Sub	Publication Type
	Acc	Sen	Spe	MAE, bpm	MAPE, %	r-Pearson			
[24], 2017	95	-	-	-	-	-		30	Research
[25], 2018	-	-	-	-	-	0.856		30	Research
[27], 2019	-	-	-	-	-	0.820		25	Research
[36], 2020	97	-	-	-	-	-		41	Commercial
[42], 2020	-	85	90	-	-	0.870		50	Research
[33], 2020	-	-	-	2.1	-	0.920		40	Commercial
[32], 2020	-	-	-	7.4	12.46	-		5	Research
[31], 2020	-	-	-	-	3.6	0.860		10	Research
[37], 2020	-	-	-	-	-	0.937		11	Research
[39], 2020	-	-	-	-	-	0.961		30	Research
[48], 2021	95	-	-	0.02	-	-	For IR-Camera algorithm	20	Research
	99						For RGB-Camera algorithm		
[49], 2021	97	98	-	-	-	-		25	Research
	89	94							
[50], 2021	75	-	-	-	-	-		22	Research
[51], 2021	97	-	-	-	3.6	-		45	Commercial
[47], 2021	93	-	-	-	1.06	0.983		6	Research
					1.15	0.987			
[43], 2021	95	-	-	-	-	0.892		40	Research
[44], 2021	91	-	-	-	-	-		20	Research
	94								
[52], 2021	-	-	-	2.39	-	-		12	Research
[46], 2021	-	-	-	1.17	-	-		11	Research
[61], 2022	86	-	-	-	-	-		10	Research
[63], 2022	97	-	-	-	-	-		24	Research
[53], 2022	-	-	-	6.7	-	-		18	Research
[66], 2022	-	-	-	2.9	-	-		463	Commercial
[57], 2022	-	-	-	1.28	1.74	-		20	Research
[58], 2022	-	-	-	4.28	5.56	-		15	Research
[56], 2022	-	-	-	3	-	-		42	Research
				8.6					
[59], 2022	-	-	-	-	-	0.998		6	Research

## Data Availability

The authors will make the datasets that support the conclusions of this article available upon reasonable request.

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
