# Peer review of "Contactless Technologies, Sensors, and Systems for Cardiac and Respiratory Measurement during Sleep: A Systematic Review"

_sensors, 2023, doi:10.3390/s23115038_

Round 1

Reviewer 1 Report

This is a much needed overview of a rapidly developing scientific area of contactless monitoring of sleep. Having been working in this area for several years and by nature being (some say too) critical, this systematic review leaves hardly anything to be critical about.

Emphasizing that my own work or that of my direct colleagues in this area has not been cited, I can't think of any point of critic. In short: congratulations on a great manuscript and overview.

Author Response

Dear Sir/Madam,

Thank you very much for your review and response!

I have revised and updated the manuscript based on the all comments from all reviewers.
Please see the attachment.

Best regards,

Andrei Boiko

Reviewer 2 Report

While I appreciate the comprehensive literature review and analysis of non-contact technologies for monitoring cardiorespiratory activity during sleep, I find that the manuscript does not meet the impact criteria of the journal. The provided information is not entirely novel and the manuscript does not make any significant contribution to the existing knowledge base. Additionally, the manuscript lacks clarity and coherence in some parts, and the writing style is not consistently clear and concise. The introduction and research questions did not receive sufficient attention. Therefore, I do not recommend publication in this journal.

The quality of English language in this manuscript requires improvement.

Author Response

Dear Sir/Madam,

Thank you very much for your response and review!

Here you can find a point-by-point response based on your comments and comments of your colleagues:

1. "The provided information is not entirely novel and the manuscript does not make any significant contribution to the existing knowledge base."

We have added in the Introduction section the information what corresponds to the actual state-of-the-art related to the topic of investigation. Based on the several mentioned references, our work perhaps actualise the existing knowedge base.

2. "The introduction and research questions did not receive sufficient attention."

Based on your comment, the Introduction section was revised and expanded. 

3.  "The quality of English language in this manuscript requires improvement."

The English in the manuscript was revised and improved.

Best regards,

Andrei Boiko

Reviewer 3 Report

The topic seems interesting however there are some concerns which should be addressed.

1.      The abstract is not very clear it should be rewritten.

2.      English should be checked throughout the manuscript.

3.      Research gap should be highlighted in the introduction, and how this manuscript will help in future research should be presented.

4.      It is crucial to have knowledge of the various methodologies used to gather Cardiac and Respiratory measurements during sleep. The working principles of the various technologies are missing. Moreover, there is no comparison of the various methodologies and their advantages and drawbacks.

5.      Some findings such as the limitations of the state-of-the art technologies, current research gap, and future directions etc. should be detailed in a separate section.

6.      A detailed conclusion will be better at the current form the conclusion is not so impressive.

English should be checked throughout the manuscript.

Author Response

Dear Sir/Madam,

Thank you very much for your response and review!

Please see the new attachment and our point-to-point answerbased on your comments:

1."The abstract is not very clear it should be rewritten."

The abstract section was revised and rewritten.

2. "English should be checked throughout the manuscript."

The English in the manuscript was revised and improved.

3. "Research gap should be highlighted in the introduction, and how this manuscript will help in future research should be presented."

We have revised and updated the Introduction section. So, we highlighted the significance of this article in future research and mentioned the research gap. Additionally, we add the our opinion and suggestions related to significance the paper in future (and possible future directions) in the Discussion and Conclusion sections.

4. "It is crucial to have knowledge of the various methodologies used to gather Cardiac and Respiratory measurements during sleep. The working principles of the various technologies are missing. Moreover, there is no comparison of the various methodologies and their advantages and drawbacks."

Based on this comment, we have added the description of the considered and existing technologies of contactless cardiorespiration monitoring in the Results section. Moreover, we compared these methodologies and their possible advantages, disadvantages and applications.

5. "Some findings such as the limitations of the state-of-the art technologies, current research gap, and future directions etc. should be detailed in a separate section."

The limitations of the state-of-the art technologies as well as future directions and research gap were written in the Results section in details.

6. "A detailed conclusion will be better at the current form the conclusion is not so impressive."

The Conclusion section was expanded based on this comments. However, it was expanded also based on the obtained results.

Best regards,

Andrei Boiko

Round 2

Reviewer 2 Report

After the modifications, the content of the article has significantly improved. Here are my suggestions:

1. In manuscript, it is mentioned that 3.3.2 SRQ-1 and 3.3.3 SRQ-2 are the same. Please check.

2. Is it possible to assess respiratory conditions by measuring the parameter of respiratory humidity in the human body?

3. Surface acoustic wave (SAW) technology enables wireless and passive remote sensing, and has been developed for sensing parameters such as temperature, humidity, and strain. For example, in the Journal of Micromechanics and Microengineering, 2017, 27(11): 115006, SAW technology is discussed as a means to monitor respiratory conditions in humans. The authors argue whether it is necessary to introduce SAW sensing technology. Relevant references are as follows: ACS Appl. Mater. Interfaces 2020, 12, 35, 39817–39825, and Nano Energy, 2020, 78: 105307.

4. What are the challenges and considerations in implementing contactless monitoring technologies for real-world clinical and home settings, and how can they be addressed to ensure accurate and reliable measurements?

Author Response

Dear Sir/Madam,

Thank you very much for your response and review!

Here you can find a point-by-point response based on your comments and comments of your colleagues:

  1. "In manuscript, it is mentioned that 3.3.2 SRQ-1 and 3.3.3 SRQ-2 are the same. Please check."

    We have added the text with the description of our accents to each of the mentioned subsections.

  2. "Is it possible to assess respiratory conditions by measuring the parameter of respiratory humidity in the human body?"

    It is technically possible.
    However, the focus of this work has been on monitoring cardiorespiratory activity during sleep, which mainly involves monitoring HR and EF. The measurement of respiratory humidity, on the other hand, is mainly related to the detection of apnoea, which (in our case) is only complementary to a system capable of measuring respiratory rate. In addition, the algorithms used to detect apnoea may in some cases differ from those used to detect respiratory rate, which we believe is a separate topic for discussion outside the scope of this review.

  3. "Surface acoustic wave (SAW) technology enables wireless and passive remote sensing, and has been developed for sensing parameters such as temperature, humidity, and strain. For example, in the Journal of Micromechanics and Microengineering, 2017, 27(11): 115006, SAW technology is discussed as a means to monitor respiratory conditions in humans. The authors argue whether it is necessary to introduce SAW sensing technology. Relevant references are as follows: ACS Appl. Mater. Interfaces 2020, 12, 35, 39817–39825, and Nano Energy, 2020, 78: 105307."

    Our systematic review focuses on non-contact technologies for measuring cardiorespiratory parameters. In other words, there is no visible contact between the patient's body and the sensor or device providing the measurement. It is clear from the articles presented, as well as those related to the above, that this is a contact technology, which excludes it from the scope of this review. However, we cannot ignore the potential of this technology.

  4. "What are the challenges and considerations in implementing contactless monitoring technologies for real-world clinical and home settings, and how can they be addressed to ensure accurate and reliable measurements?"

    We have added the text with the probable challenges and considerations related to this comment in the subsection 3.3.6 and Conclusion section. 

Best regards,

Andrei Boiko

Reviewer 3 Report

All my queries have been addressed.

Author Response

Dear Sir/Madam,

Thank you very much for your review and response!

I have revised and updated the manuscript based on the all comments from all reviewers.

Best regards,

Andrei Boiko